# Identification of the Determinants of Plexiform Neurofibroma Morbidity in Pediatric and Young Adult Neurofibromatosis Type 1 Patients: A Pilot Multivariate Approach

**DOI:** 10.3390/cancers17010123

**Published:** 2025-01-02

**Authors:** Biagio de Brons, Britt Dhaenens, Rick van Minkelen, Rianne Oostenbrink

**Affiliations:** 1Sophia, Department of General Pediatrics, Erasmus MC, 3015 GD Rotterdam, The Netherlands; 445070bb@student.eur.nl (B.d.B.);; 2Department Genetics, Erasmus MC, 3015 GD Rotterdam, The Netherlands; r.vanminkelen@erasmusmc.nl; 3The ENCORE Expertise Centre for Neurodevelopmental Disorders, Erasmus MC, 3015 GD Rotterdam, The Netherlands

**Keywords:** longitudinal history, plexiform neurofibroma, predictors, neurofibromatosis type 1

## Abstract

We studied a group of 90 patients with NF1-related plexiform neurofibromas (PNs) with an average age of 15.7 years and a follow-up period of 9.8 yrs. PNs in patients with older age and PNs located in the craniofacial region or trunk were most frequently associated with plexiform neurofibroma morbidity and most frequently underwent intervention (surgery or systemic treatment). These findings may contribute to decisions on whether or not to start what treatment in NF1 patients with PNs.

## 1. Introduction

Neurofibromatosis type 1 (NF1), although rare, with an incidence of approximately 1 in 2700, is one of the most frequent tumor-suppressor multisystem syndromes [1,2]. NF1 is characterized by a wide variability in manifestations and severity among patients. These include skin abnormalities (café au lait macules, axillary or inguinal freckling), eye abnormalities, musculoskeletal manifestations (scoliosis, pseudoarthrosis), nervous system tumors (neurofibromas, optic pathway gliomas (OPGs)), epilepsy, vascular complications, and cognitive problems [3,4]. NF1 is caused by a germline pathogenic variant in *NF1*, consisting of 57 exons (transcript NM_000267.3) that span over 350 kb of genomic DNA, located on chromosome 17q11.2 [5]. *NF1* encodes neurofibromin, a protein that serves as a negative regulator of the RAS/MAPK pathway [6]. This pathway plays a key role in cell growth, division, and differentiation. Therefore, the *NF1* gene functions as a tumor-suppressor gene 2.

A major complication in NF1 is plexiform neurofibroma (PN), occurring in 30% (visible) to 50% (on imaging) of NF1 patients [3,7]. PNs are present in principle from birth but may grow and develop into large benign tumors along large nerves in the body [8]. Due to their location or growth, they may cause neurological deficit, pain, or the obstruction of vital organs. A major risk of PN is the transformation into a malignant peripheral nerve sheath tumor (MPNST), with a 10% lifetime risk in patients with NF1, representing a major cause of mortality in NF1 patients [9,10]. Until the past decade, treatment was limited to surgery, which in many cases does not achieve complete tumor resection. Surgery also has a considerable risk of postoperative deficits due to the large size and the invasion of the surrounding structures of the PN [11]. Furthermore, long-term benefits of surgery are unlikely in patients under the age of 10 with craniofacial or trunk lesions, mainly because the resection is often incomplete. In 2020, selumetinib, an MEK inhibitor, was approved by the FDA (by EMA in 2021) for the treatment of children with inoperable symptomatic PNs [12,13].

Currently, it is unclear which NF1 patients with PNs are at risk for disease progression and need which therapy at which timepoint. The scientific community is trying to identify new outcomes to predict the course of PN using anatomical, radiological, and biological characteristics to identify PNs with a high risk for progression [14,15]. The scarcity of natural history studies on PNs limits our understanding of their natural progression. Natural history study data are imperative to uncover patient characteristics that can serve as predictors for the development of symptomatic or fast-growing PNs. Therefore, this study aims to outline the characteristics of NF1 patients with PN and compare those at high risk for PN progression or experiencing significant morbidity from PN with individuals who have lower morbidity or risk levels.

## 2. Materials and Methods

### 2.1. Study Design and Patients

This is a retrospective cohort study. We included patients currently alive with a diagnosis of PN and NF1, based on the revised diagnostic criteria for neurofibromatosis type 1 [16]. The patients were seen between 2012 and 2023 at the ENCORE-NF1 expertise center in the Erasmus MC-Sophia Children’s Hospital in the Netherlands. Patients and/or their caregivers provided informed consent for the use of medical record data for research (local Institutional Review Board identifier MEC-2015-203).

### 2.2. Outcomes

The primary outcome of interest was the occurrence of a plexiform neurofibroma at risk of or causing morbidity, further referred to as ‘plexiform neurofibroma morbidity’, and was defined as a PN with progressive growth, a PN that causes morbidity (excluding disfigurement/cosmetic burden), or a PN that causes pain needing intervention (analgesics, or warranting systemic or surgical treatment). The secondary outcome measure was the age at first therapeutic intervention (surgery or MEK-inhibitor treatment) of the PN where therapy was targeted for the patient.

### 2.3. Data

We collected data from medical records, including biological sex, date of birth, age at NF1 diagnosis, date of first and last evaluation, inheritance (familial or de novo), type of NF1 pathogenic variant, and diagnostic criteria associated with NF1 [16]. At the PN level, we gathered data on location, date of diagnosis, and available imaging data. PN location was categorized as either trunk, craniofacial, or limb. At the patient level, we gathered data on the number of PNs and their morbidity and size measurement. Volumetric analysis was not performed routinely in our center. As bidimensional reported sizes do not correspond well to tumor volume [17], we did not include the (change in) size in further analyses. Comorbidities related to PNs were categorized as disfigurement and/or cosmetic burden, pain, neurological deficit, airway complications, vision impairment, bowel or bladder complications, and others. Details of interventions included the type of intervention (surgical or systemic) and its date.

### 2.4. Statistical Analyses

Descriptive statistics were computed for primary outcomes and determinants, including means and standard deviations for continuous variables and frequencies for categorical variables. Continuous variables were analyzed using Mann–Whitney U tests, while chi-squared or Fisher’s exact tests were performed for categorical variables. Patients with MPNST were excluded from this analysis and described separately. Determinants for complex PN were assessed by multivariate logistic regression analysis. In addition, we assessed determinants for the timing of the intervention by Cox regression. To prevent overfitting for both logistic regression analyses, given the expected low number of cases, we pre-selected potential predictors from the literature and clinical experience in order to keep the number of variables used in both regression models about equal to one variable per ten events [18]. All calculations were performed using SPSS Predictive Analytics Software Version 29.0.2.0 (IBM Corp, Armonk, NY, USA). If not elsewhere specified, *p*-values were calculated at a 95% confidence interval.

## 3. Results

### 3.1. Demographics and Clinical Characteristics

Ninety patients were identified with a benign PN, of which 54.4% were male, with a median age at last evaluation of 15.7 years. The median age of NF1 diagnosis was 1.0 years (Table 1). The median follow-up time was 9.8 yrs. All patients fulfilled the NF1 criteria. The most frequent diagnostic criteria were six or more café-au-lait macules (100%), freckling in the axillary or inguinal regions (98%), and a detected pathogenic NF1 variant (89%). NF1-related bone deformities were present in nine patients (10%), all with sphenoid wing dysplasia, combined with pseudoarthrosis in two. Inheritance was mostly de novo (66%). Pathogenic variants were mostly nonsense/frameshift/splicing (81%), with a small proportion of our patients having NF1 microdeletion (4.8%). Twelve patients (13.3%) harbored an optic pathway glioma (OPG), which in four patients was combined with an orbital or periorbital plexiform neurofibroma (OPPN).

In addition to this cohort, we observed four patients who developed an MPNST in the same study period, occurring at a median age of 10 years (one child below 5 years and one child above 15 years). Genetic variant types were two nonsense/frameshift, one missense, and one microdeletion. The MPNSTs were in the cervical/thoracic region (*n* = 2), upper limb, and abdomen.

In the 90 patients with a benign PN present, the limbs were the predominant site (Table 2). Fourteen patients (15.5%) of the population had more than one PN. Forty-three patients experienced comorbidities associated with their PN, predominantly pain requiring intervention and disfigurement. Based on the earlier defined criteria, 37 patients had plexiform neurofibroma morbidity. Twenty-eight patients received an intervention: 19 underwent resection at a median age of 13.4 years; 3 were primarily treated with an MEK inhibitor (median age 16.4 yrs), which have been accessible in the Netherlands through a compassionate use program since 2021; and 6 were treated with MEK as an adjuvant therapy to previous surgery at a median age of 5.2 yrs.

### 3.2. Predictors of Complexity and Need for Intervention

At the patient level, we observed no differences in univariate analysis for age at last evaluation, follow-up duration, or distribution of sex for the development of plexiform neurofibroma morbidity (Table 3), nor in any diagnostic criteria or genetic mutation type. Trunk PNs had a higher plexiform neurofibroma morbidity in comparison to limb PNs, although not significant. Within the craniofacial PNs, we observed 11 OPPNs; 10 of them were related to plexiform neurofibroma morbidity (27%), versus only 1 (1.9%) in an individual without plexiform neurofibroma morbidity (*p* < 0.001).

An analysis of independent predictors for PNs with morbidity could be performed in 83 cases (PN with morbidity *N* = 32) due to missing mutation type data for seven patients (Table 4). We considered clinically relevant variables, including age and gender (as general patient descriptors), variant type, and location [3,9,14,19]. As a first step, we modeled patient characteristics, and in the second step, we added PN location to the model. Older age was significantly related to the risk of plexiform neurofibroma morbidity, independent of variant type or location. This model accounted for 37.0% of the variance in the occurrence of complex PN. It demonstrated an overall predictive accuracy of 68.7%, correctly identifying 92.2% of PN cases without morbidity and 31.3% of PN cases with morbidity. The model’s discriminative ability for PNs with morbidity was limited (area under the ROC curve (AUC) of 0.64 (95% CI, 0.52 to 0.76)). Regarding independently associated determinants for the need for PN intervention in 81 patients with complete data (patients with intervention *n* = 25), Cox regression identified the location of the PN, with craniofacial region as the strongest predictor (Table 5). The mutation type did not contribute further.

## 4. Discussion

### 4.1. Main Findings

Among a pediatric population of 90 NF1 patients with PNs, 37 developed plexiform neurofibroma morbidity over a median follow-up duration of 10 years. Fifteen percent of the patients developed multiple PNs. Older age was related to plexiform neurofibroma complexity, independent of genetic variant type or location. In addition, craniofacial PNs were independently associated with the early need for intervention.

Comparing our outcomes with findings in the literature revealed that our population was mostly similar to the general NF1 population [20,21,22]. The age of diagnosis was based on retrospective reports and may be biased by the first moment of consultation, so it is difficult to compare with other cohorts. The low incidence of MPNST (5%) may be due to the relatively young patient population.

The higher need for intervention for plexiform neurofibroma morbidity in craniofacial and trunk PNs can be explained by their potential morbidity. The craniofacial location may yield more complaints due to its limited anatomical space, where vital structures are closely packed. This is supported by the fact that 73% of morbidity in our OPPN group was due to visual impairment. Studies report visual impairment to be caused by conditions such as congenital glaucoma and/or amblyopia, which may result from eyelid closure caused by severe ptosis due to a PN, often in combination with an associated OPG [23]. Notably, four of our cases involved both OPPN and OPG simultaneously. Early intervention may be applied to improve and/or preserve functional ability and cosmetic outcomes. Systemic treatment with an MEKi is positioned for the treatment of OPPN [13]. Trunk PNs (including spinal, thoracic, and abdominal PNs) may be considered to threaten vital structures and warrant earlier intervention before causing actual symptoms (this was included in our definition of plexiform neurofibroma morbidity). This explains their association with a higher probability of intervention compared to that for limb PNs. The limited number of cases did not allow us to assess functionally relevant sub-locations to the main anatomical sites, such as vision loss, hearing loss, spinal cord compression, airway compromise, bowel/bladder disfunction, or mobility. We were unable to identify a microdeletion of the NF1 gene as a determinant in the occurrence of plexiform neurofibroma morbidity, while current literature suggests an association between NF1 microdeletions and an increased incidence of high tumor burden in PNs [9,24]. However, this may be related to the small number of patients with microdeletions in our population, limiting the power of our comparison.

### 4.2. Strengths and Limitations

A strength of this study is that we were able to include a high number of children with NF1-PN, with a median follow-up of 10 years. The ENCORE-NF1 expertise center Erasmus MC is the only expertise center in the Netherlands and leads the national NF1 expertise network with 11 treatment centers. In this network, complex cases are referred to our center. In addition, the Erasmus MC provides integrated NF1 care for the southern and southwestern regions of the Netherlands, which accounts for a 25% country adherence area. So, we believe that our patients are representative of the general NF1 population in the Netherlands [25]. Next, we add to previous longitudinal studies on NF1-related PNs [8,19,20,26] regarding the potential roles of demographic and PN characteristics in NF1 patients in developing plexiform neurofibroma morbidity and a need for intervention. Our study also has several limitations. First, although we employed a substantially large cohort, we still had a small number of cases of plexiform neurofibroma morbidity and PN with intervention. This limited the number of variables included in our regression analyses, and the models may be subject to overfitting. To reduce overfitting, we applied an approach of preselecting potential determinants from the literature. This, however, limited the chance of new findings. Second, the study is based on routine care data, and thereby subjective to changes in clinical management over time. Not all data were available for all patients. Also, we do not include volumetry measurements from MRI in routine care, limiting the use of (change in) tumor volume as a predictor in this study. In addition, we did not have standardized MRI intervals but performed MRI on indication (patients showing symptoms or suspected growth that warranted potential intervention). Monitoring PN growth (in absence of symptoms) was not an indication for MRI in the period before MEK inhibitors became available. Next, intervention was decided by the discretion of the clinician but included the level of complaints (deficit, pain), potential threats to vital organ structure/function, growth, and patient preferences (despite the latter being rather subjective and potential variable). Treatment decisions, however, were made by the multidisciplinary team, including the neurology, ophthalmology, neurosurgery, and plastic surgery departments, the members of which have been quite stable in the past 10 years. The access to systemic treatment with MEK inhibitors since 2021 may have influenced the time to treatment for inoperable PNs. The retrospective nature forced us to use objective definitions of parameters. This caused us to decide to exclude disfigurement/cosmetic burden in our definition of plexiform neurofibroma morbidity, but these aspects can also have a significant impact on quality of life and may benefit from treatment. For future prospective longitudinal studies, we underline the importance of including functional and patient-reported outcomes, which are becoming increasingly available for NF1 specific manifestations [27,28,29].

### 4.3. Future Directions

Our study has provided valuable insights into NF1-related PN morbidity. Our results support initiating the treatment of PNs early in life, for craniofacial PNs in particular. The choice of surgery versus systemic MEK inhibitor treatment may depend on PN characteristics, accessibility, and experience and should be discussed in multidisciplinary teams. Of course, there remain several challenges for further research to enhance our understanding of PN natural progression. Some of our study limitations can be overcome by a prospective multicenter LNHS rather than selective patient populations from clinical trials or smaller observational cohorts from local institutions, as in the current study. This, however, is more financially and temporally demanding. Such a prospective approach also allows researchers to collect patient-derived material for new therapeutic targets and biomarkers for NF1-PN progression or transformation into MPNST [30,31,32]. There are some efforts to build networks for NF1, aiming to cover the general population of NF1 patients, e.g., the CTF clinical care network in the USA (https://www.ctf.org/nf-clinic-network/, accessed on 10 September 2024) and ERN-GENTURIS [33] in Europe, but not all NF1 patients have access to network centers. Next, a systematic data collection procedure with standardized protocols among network centers is needed, such that observational data can be harmonized and combined for studying rare outcomes in rare conditions. REiNS provide valuable recommendations for both imaging and functional or patient-reported outcome assessments [27], but these are not yet fully implemented in routine care. Integrating volumetric analysis holds promise for the more accurate and earlier detection of complex PNs, as well as the improved assessment of treatment response, but adopting this in routine practice requires (semi)automatic analyses [34]. Collaborators from REiNS and the EU-PEARL consortium have proposed content for longitudinal studies, but these are not broadly implemented yet [32,35].

## 5. Conclusions

Our pilot multivariate approach provides insights into the natural course of plexiform neurofibromas in neurofibromatosis type 1 patients and into potential factors independently influencing plexiform neurofibroma morbidity and the need for intervention. Our results support initiating the treatment of PNs early in life before complications become present. In addition, the location of the PN may contribute to the treatment decisions in NF1 patients. Harmonized approaches to follow-up on PNs among global NF centers are needed, as they will allow researchers to bundle longitudinal natural history cohorts. Such large multicenter cohorts may confirm our findings, identify additional prognostic determinants, accelerate our understanding of the PN course, and advance PN management strategies.

## Figures and Tables

**Table 1 cancers-17-00123-t001:** Demographic and clinical characteristics of patients with benign PNs.

Characteristic	Study Population (*N* = 90)
**Age at last evaluation, years, median (range)**	15.7 (3.3–28.6)
**Duration of follow-up, years, median (range)**	9.8 (0–26.4)
	***N* = 76**
**Age at NF1 diagnosis, years, median (range)**	1.0 (0–14)
**Sex, *N* (%)**	***N* = 90**
Male	49 (54.4)
**Inheritance, *N* (%)**	***N* = 88**
Familial	31 (34.4)
De novo	57 (63.3)
Unknown	2 (2.2)
**Variant type, *N* (%) ***	***N* = 83**
Missense/in-frame deletion	11 (13.3)
Nonsense/frameshift/splicing	67 (80.7)
Intragenic deletion	1 (1.2)
Total NF1 deletion (microdeletion)	4 (4.8)

* Known pathogenic variants based on the LOVD classification in 83 patients (www.lovd.nl, access date 10 September 2024).

**Table 2 cancers-17-00123-t002:** Characteristics of plexiform neurofibromas.

Characteristics of PNs	At the PN Level (*N* = 105)	At the Patient Level (*N* = 90)
**Location, *N* (%)**		
Craniofacial	36 (34.29)	
Trunk	26 (24.76)	NA
Limbs	43 (40.95)	NA
**Number of PNs per person, *N* (%)**		
1	NA	76 (84.4)
2	NA	13 (14.4)
3	NA	1 (1.1)
**PN morbidity, *N* (%) ***		***N* = 48**
Pain requiring intervention	23 (21.7)	21 (43.8)
Disfigurement	16 (15.1)	16 (33.3)
Neurological deficit	6 (5.7)	6 (12.5)
Visual impairment **	9 (8.5)	9 (18.8)
ENT-related complaints	3 (2.8)	3 (6.2)
Ptosis	3 (2.8)	3 (6.2)
Skeletal complaints	3 (2.8)	3 (6.2)
Sensory complaints	2 (1.9)	2 (4.2)
Hemorrhage in the PN	1 (0.9)	1 (2.1)
Urogenital complaints	1 (0.9)	1 (2.1)
**Intervention, *N* (median age (yrs) of first intervention)**		28 (11.4)
Surgery	x	19 (13.4)
MEK inhibitor	x	3 (16.4)
Both	x	6 (5.2)

NA not applicable; * more than one comorbidity can be related to one PN; ** in all cases caused by an OPPN.

**Table 3 cancers-17-00123-t003:** Univariate comparison of characteristics between patients with and without plexiform neurofibroma morbidity.

Characteristic	Plexiform Neurofibroma Without Morbidity (*N* = 53)	Plexiform Neurofibroma With Morbidity (*N* = 37)	*p* Value Category *	*p* Value *
Age at last evaluation, years, median (range)	15.7 (3.3–27.7)	15.9 (6.5–28.6)		0.36
Follow-up duration, years, median (range)	8.2 (0–23.8)	10.9 (0–3.4)		0.09
**Sex, *N* (%)**				
Male	30 (56.6)	19 (51.4)		0.63
**Inheritance, *N* (%)**	***N* = 52**	***N* = 36**		
Familial	19 (36.5)	12 (33.3)		
De novo	33 (63.5)	24 (66.6)		
**Variant type, *N* (%) ****	***N* = 51**	***N* = 32**	0.84	
Missense/in-frame deletion	6 (11.8)	4 (12.5)		
Nonsense/frameshift/splicing	42 (82.4)	26 (81.3)		
Total NF1 deletion (microdeletion)	2 (3.9)	2 (6.3)		
Intragenic deletion	1 (2.0)	0		
**Location of target PN, *N* (%)**	***N* = 53**	***N* = 37**	0.41	
Craniofacial	20 (37.7)	14 (37.8)		
Trunk	10 (18.9)	11 (29.7)		
Limbs	23 (43.4)	12 (32.4)		

* Statistical significance at *p* < 0.0045; ** missing in seven patients.

**Table 4 cancers-17-00123-t004:** Multivariate logistic regression model of determinants for plexiform neurofibroma morbidity, adjusted for age at last evaluation and biological sex.

	Model 1		Model 2	
Variable	Odds Ratio (95% CI)	*p* Value	Odds Ratio (95% CI)	*p* Value
**Age at last evaluation**	1.05 (0.98–1.13)	0.16	1.05 (0.98–1.13)	0.05
**Biological sex ***	0.75 (0.30–1.86)	0.53	0.65 (0.25–1.69)	0.38
**Variant type ****				0.96
Nonsense/frameshift/splicing	1.02 (0.26–4.03)	0.98	0.86 (0.21–3.53)	0.83
Total NF1 deletion (microdeletion)	1.85 (0.17–20.70)	0.62	1.47 (0.3–17.2)	0.76
Intragenic deletion	0.000(0–∞)	1.00	0.000 (0–∞)	1.000
**Location of target PN *****				0.41
Trunk	x	x	2.32 (0.68–7.96)	0.18
Craniofacial	x	x	1.34 (0.45–3.97)	0.60
**AUC (95% CI)**	0.62 (0.49–0.74)		0.64 (0.52–0.76)	

* Reference category: female; ** reference category: missense/in-frame deletion; *** reference category: limb PN.

**Table 5 cancers-17-00123-t005:** Cox regression model of determinants for PN intervention (*n* = 81 patients with complete data, *n* = 25 patients with intervention).

	Coefficient (S.E.)	Hazard Ratio (95% CI)	*p* Value
Biological sex *	0.51 (0.43)	1.67 (0.72–3.89)	0.24
**Location of target PN** **			0.08
Trunk	1.14 (0.65)	3.13 (0.88–11.17)	0.08
Craniofacial	1.42 (0.58)	4.15 (1.34–12.86)	0.01
**Variant type** ***			0.61
Nonsense/frameshift/splicing	−0.22 (0.64)	0.80 (0.23–2.81)	0.73
Total NF1 deletion (Microdeletion)	0.75(0.96)	2.13 (0.33–13.84)	0.43
Intragenic deletion	−11.70 (645.71)	0.00 (0.00–∞)	0.99

* Reference category: female; ** reference category: limb PN; *** reference category: missense/in-frame deletion.

## Data Availability

Data are available on request from the corresponding author. Data are available as .csv formats.

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
