# Peer review of "Identification of the Determinants of Plexiform Neurofibroma Morbidity in Pediatric and Young Adult Neurofibromatosis Type 1 Patients: A Pilot Multivariate Approach"

_cancers, 2025, doi:10.3390/cancers17010123_

Round 1

Reviewer 1 Report

Comments and Suggestions for Authors

The study of Biagio de Brons et al. is part of a panorama of great interest in studying tumours arising in neurofibromatosis, such as plexiform neurofibromas (PN). These tumours, probably congenital, grow unpredictably during childhood, adolescence and adulthood. The scientific community is trying to identify new outcomes to predict the course of PN (anatomical characteristics [Kathryn M Lemberg et al. Pediatr Res 2024 Aug 28. doi: 10.1038/s41390-024-03474-z] appearance on MRI [Schmalhofer et al. Orphanet Journal of Rare Diseases (2024) 19:412 https://doi.org/10.1186/s13023-024-03420-6] and biological complexity) to identify PN with a high risk for progression. Specifically, the study aims to assess the phenotype determinants in PN as predictive features of PN complexity and timing of intervention for PN.

Thus, the authors retrospectively compare the characteristics of NF1 patients with PN and those at high risk for PN progression (complex PN) with those with PN of lower complexity.

To this aim, the authors analysed a cohort study on clinical data from patient records of NF1 patients with PNs seen at the Sophia Children’s Hospital in the Netherlands between 2012 and 2023. Since thirty-seven out of 90 patients bearing benign PN, developed complex PNs during follow-up, the author concluded that older age is associated with higher PN complexity and can considered as a predictive factor of PN progression.

 What the authors found is really interesting and clinically relevant. They showed that older age and location were significantly associated with higher PN complexity; remarkably, they found the strongest association for orbital/periorbital PN (OPPN) predictive of careful assessment for prompt therapeutic intervention as surgery.

We consider this work essential for the hypothesis proposed and the clinical implications it could have in the care and follow-up of PN patients. The number of patients analyzed is adequate, the analyses are statistically relevant, and the conclusions are supported overall. 

Minor point:

Add the following references into the introduction for completness:

 - Kathryn M Lemberg et al. Pediatr Res 2024 Aug 28. doi: 10.1038/s41390-024-03474-z

  - Schmalhofer et al. Orphanet Journal of Rare Diseases (2024) 19:412    https://doi.org/10.1186/s13023-024-03420-6

Author Response

Comment 1. The study of Biagio de Brons et al. is part of a panorama of great interest in studying tumours arising in neurofibromatosis, such as plexiform neurofibromas (PN). These tumours, probably congenital, grow unpredictably during childhood, adolescence and adulthood. The scientific community is trying to identify new outcomes to predict the course of PN (anatomical characteristics [Kathryn M Lemberg et al. Pediatr Res 2024 Aug 28. doi: 10.1038/s41390-024-03474-z] appearance on MRI [Schmalhofer et al. Orphanet Journal of Rare Diseases (2024) 19:412 https://doi.org/10.1186/s13023-024-03420-6] and biological complexity) to identify PN with a high risk for progression. Specifically, the study aims to assess the phenotype determinants in PN as predictive features of PN complexity and timing of intervention for PN.

Thus, the authors retrospectively compare the characteristics of NF1 patients with PN and those at high risk for PN progression (complex PN) with those with PN of lower complexity.

To this aim, the authors analysed a cohort study on clinical data from patient records of NF1 patients with PNs seen at the Sophia Children’s Hospital in the Netherlands between 2012 and 2023. Since thirty-seven out of 90 patients bearing benign PN, developed complex PNs during follow-up, the author concluded that older age is associated with higher PN complexity and can considered as a predictive factor of PN progression.

 What the authors found is really interesting and clinically relevant. They showed that older age and location were significantly associated with higher PN complexity; remarkably, they found the strongest association for orbital/periorbital PN (OPPN) predictive of careful assessment for prompt therapeutic intervention as surgery.

We consider this work essential for the hypothesis proposed and the clinical implications it could have in the care and follow-up of PN patients. The number of patients analyzed is adequate, the analyses are statistically relevant, and the conclusions are supported overall. 

Reply 1: Thanks for underlying the importance of our findings.

Comment 2. Minor point:

Add the following references into the introduction for completness:

 - Kathryn M Lemberg et al. Pediatr Res 2024 Aug 28. doi: 10.1038/s41390-024-03474-z

  - Schmalhofer et al. Orphanet Journal of Rare Diseases (2024) 19:412    https://doi.org/10.1186/s13023-024-03420-6

Reply 2: Thanks for the suggestions, which information and references have been added to the last paragraph of introduction (line 64-66).

Reviewer 2 Report

Comments and Suggestions for Authors

The authors analyze patient and plexiform neurofibroma (PN) characteristics that contribute to PN morbidity and the need for treatment interventions in patients with neurofibromatosis type 1 (NF1). 

I question whether introducing the new term of “complex PN” is the best way to present the findings. In the context of NF1, complex PN is already used to describe the shape/structure of these tumors that may lead to confusion.

The definition of complex PN in the manuscript is:  a PN with (1) progressive growth, a PN that (2) causes morbidity (excluding disfigurement/cosmetic burden), or a PN that (3) causes pain needing intervention (analgesics, or warranting systemic or surgical treatment).  However, later in the text (line 93) the authors state that size change in not included in the PN characteristics, as neither 3D nor 2D measurements were not consistently available.  That leaves morbidity (with the exclusion of disfigurement and inclusion of pain requiring treatment) the essence of the new “complex PN”.   I recommend changing the complex PN category to PN with morbidity, and the title to:   Determinants of Plexiform Neurofibroma Morbidity in Pediatric and Young Adult NF1 Patients…

The most prominent conclusion of the article is that orbital/periorbital PN location is associated with increased risk of morbidity and higher hazard of intervention.  PN locations were divided into 4 categories: (1) orbital/periorbital, (2) craniofacial, (3) trunk, and (4) limb.  These categories are somewhat arbitrary and seem to be selected with the potential for morbidity already considered.  Separating orbital/periorbital PN as an independent category (and not as a sub-group of craniofacial PN, where it actually belongs) strongly suggest that the authors are aware that it is a high-risk location and therefore make a circular argument.  Limiting the anatomical sites to 3 (head/neck, trunk, and limbs), and adding functionally relevant sub-locations to the main categories that account for the potential for hearing loss, spinal cord compression, airway compromise, bowel/bladder disfunction, mobility, etc. would allow an unbiased assessment.    

The 4 patients with MPNST, while interesting, do not factor into any of the analyses, but confuse the descriptive statistics.  The text refers to 94 patients and gives discrepant values from table 1 that includes 90 patients (male 51.6% vs 54.4%; age at NF1 diagnosis 1.5 years vs 1 year).  As the topic of the article is complex PN (or PN with morbidity), I would consider excluding the MPNST cases and limiting the paper to 90 subjects with benign PN.

One minor comment: Combined OPPN and OPG mentioned in 4 patients under results (line 119) and 3 in discussion (line 180).

Author Response

Comment 1. I question whether introducing the new term of “complex PN” is the best way to present the findings. In the context of NF1, complex PN is already used to describe the shape/structure of these tumors that may lead to confusion.

The definition of complex PN in the manuscript is:  a PN with (1) progressive growth, a PN that (2) causes morbidity (excluding disfigurement/cosmetic burden), or a PN that (3) causes pain needing intervention (analgesics, or warranting systemic or surgical treatment).  However, later in the text (line 93) the authors state that size change in not included in the PN characteristics, as neither 3D nor 2D measurements were not consistently available.  That leaves morbidity (with the exclusion of disfigurement and inclusion of pain requiring treatment) the essence of the new “complex PN”.   I recommend changing the complex PN category to PN with morbidity, and the title to:   Determinants of Plexiform Neurofibroma Morbidity in Pediatric and Young Adult NF1 Patients…

Reply 1: Thank you for this comment. We indeed struggled with appropriate terminology and agree with potential overlap with complexity in terms of shape. We copied your suggestion throughout the paper.

Comment 2. The most prominent conclusion of the article is that orbital/periorbital PN location is associated with increased risk of morbidity and higher hazard of intervention.  PN locations were divided into 4 categories: (1) orbital/periorbital, (2) craniofacial, (3) trunk, and (4) limb.  These categories are somewhat arbitrary and seem to be selected with the potential for morbidity already considered.  Separating orbital/periorbital PN as an independent category (and not as a sub-group of craniofacial PN, where it actually belongs) strongly suggest that the authors are aware that it is a high-risk location and therefore make a circular argument.  Limiting the anatomical sites to 3 (head/neck, trunk, and limbs), and adding functionally relevant sub-locations to the main categories that account for the potential for hearing loss, spinal cord compression, airway compromise, bowel/bladder disfunction, mobility, etc. would allow an unbiased assessment.    

Reply 2: Thank you for notifying potential ‘circle-reasoning’. We therefore changed the analysis, and used location ‘craniofacial/trunk/limb’ instead. We confirm the value of more detailed analysis of location on morbidity as reason for intervention, but the proposed subcategories including vision, hearing, pain etc of the reviewer go beyond the power of our study, with 32 cases. We added a line on this in the discussion (line 197-199)

Comment 3. The 4 patients with MPNST, while interesting, do not factor into any of the analyses, but confuse the descriptive statistics.  The text refers to 94 patients and gives discrepant values from table 1 that includes 90 patients (male 51.6% vs 54.4%; age at NF1 diagnosis 1.5 years vs 1 year).  As the topic of the article is complex PN (or PN with morbidity), I would consider excluding the MPNST cases and limiting the paper to 90 subjects with benign PN.

Reply 3: In reply to the reviewers comment we indeed left out the MPNST cases in the figures and from the lay summary and abstract. We added however, a line with this information in the results (line 124-127), as readers may question the type of cohort if we do not report on any malignant cases.

Comment 4. One minor comment: Combined OPPN and OPG mentioned in 4 patients under results (line 119) and 3 in discussion (line 180).

 Reply 4. We corrected contradicting numbers, thanks for notifying this.

Reviewer 3 Report

Comments and Suggestions for Authors

I commend the author on their well-performed study.

I have several comments that need to be addressed before publication. 

First, how did the authors select variables for the multivariate model? I would suggest using methods like backward selection using AIC.

Please discuss the clinical significance of these findings and how the practice should be altered for these patients.

The use of time to need of intervention would be more meaningful if you only used patients who underwent surgery. Also please discuss the criteria which was used for selecting patients for treatment. 

Author Response

I have several comments that need to be addressed before publication. 

Comment 1. First, how did the authors select variables for the multivariate model? I would suggest using methods like backward selection using AIC.

Reply 1: We agree with the reviewer that this approach (backward selection) would be preferred, but this also asks for large number of cases. As we had only 32 cases, we decided to limit the set of variables in the model using clinical relevance based on literature to select these variables. We identified 4, that were present in our data, but two of them having more categories, so there is still risk of overfitting. Therefore, we did not further selection on p-values, but presented the whole model and discuss the potential contribution of the described determinants. Of course this approach inhibits new findings, but we feel that our studypopulation/size is not powered for this. We explained our analytic approach more clearly in methods (line 104-106). We made a comment on potential missed predictors in the revised version of the discussion (line 233-234).

Comment 2. Please discuss the clinical significance of these findings and how the practice should be altered for these patients.

Reply 2:  We have added some discussion on our findings to clinical practice in paragraph 4.3 future directions.

Comment 3. The use of time to need of intervention would be more meaningful if you only used patients who underwent surgery. Also please discuss the criteria which was used for selecting patients for treatment. 

Reply 3: There were only 2 patients in the analyses with systemic treatment without previous surgery, so the conclusions do not change (subanalysis available on request). However, from the clinical perspective, we think that first a decision should be made when to start treatment/to surgically intervene, and thereafter to decide upon the most appropriate treatment. Treatment choices (surgery or selumetinib) were made by discretion of the clinician, but included the level of complaints (deficit, pain), potential threat of vital structures/function, growth and the patient’s preferences (the latter being rather subjective and potential variable). This information has been added to the discussion section (line 224-224).

Round 2

Reviewer 2 Report

Comments and Suggestions for Authors

All my concerns were appropriately addressed with the revision.  I have no further comments.

Author Response

Comment 1: all my concerns were appropriately addressed with the revision.  I have no further comments.

Reply 1: thank you for reviewing and accepting the revised version as it is

Reviewer 3 Report

Comments and Suggestions for Authors

Given the limitations in addressing my prior comments, I don't believe the manuscript is suitable for publication. No added value is found in the results and several major methodological issues could not be addressed given their small sample size.

Author Response

Comment 1. Given the limitations in addressing my prior comments, I don't believe the manuscript is suitable for publication. No added value is found in the results and several major methodological issues could not be addressed given their small sample size.

Reply 1. in response to the comments we tuned down the position of the paper, by stating the pilot approach, and need for validation of observations by larger cohort multicenter studies. these changes can be found at abstract conclusion, strenghs and limitation section (line 217-219), future direction (line 246) and conclusion (line 262 and 266-269).

we changed the title accordingly.